# Joint Active Feature Acquisition and Classification with Variable-Size Set Encoding

**Hajin Shim**[1]**, Sung Ju Hwang**[1,2]**, Eunho Yang**[1,2]
KAIST[1], AItrics[2], South Korea
{shimazing, sjhwang82, eunhoy} @kaist.ac.kr

## Abstract

We consider the problem of *active feature acquisition* where the goal is to sequentially select the subset of features in order to achieve the maximum prediction performance in the most cost-effective way at test time. In this work, we formulate this active feature acquisition as a joint learning problem of training both the classifier (environment) and the *reinforcement learning* (RL) agent that decides either to 'stop and predict' or 'collect a new feature' at test time, in a cost-sensitive manner. We also introduce a novel encoding scheme to represent acquired subsets of features by proposing an order-invariant set encoding at the feature level, which also significantly reduces the search space for our agent. We evaluate our model on a carefully designed synthetic dataset for the active feature acquisition as well as several medical datasets. Our framework shows meaningful feature acquisition process for diagnosis that complies with human knowledge, and outperforms all baselines in terms of prediction performance as well as feature acquisition cost.

## 1 Introduction

Deep learning has shown remarkable growth in recent years mainly due to easier access to vast amount of data from the internet, and demonstrated significant improvements over classical and standard algorithms on diverse tasks such as visual recognition [1, 2] and machine translation [3], to name a few. The fundamental assumption for training accurate deep networks is that data is readily available at little or even no cost, such that the model can make predictions *after* observing all available features (in other words, feature acquisition is considered as an independent process against predictions). However, in some applications, information acquisition is not only affected by the model (and vice versa) but it also incurs a cost. Consider, for instance, the task of diagnosing a patient with diseases. A human doctor, in this case, will start the diagnosis by starting with only a few symptoms that the patient initially reported. From there, the doctor will either ask about more symptoms or conduct some medical examinations to narrow down the set of potential diseases the patient might have until he or she has enough confidence to make the final diagnosis. Acquiring all features (via all medical tests) in this problem may cause a financial burden to patients and more seriously it may increase the risk of not receiving proper treatment at the right time. Furthermore, collecting irrelevant features might add only noise and make the prediction unstable.

In this paper, we first formulate the feature acquisition problem that minimizes the prediction error as well as the feature acquisition cost as an optimization problem.We then provide the sequential feature acquisition framework with the classifier for predictions and the RL agent for feature acquisitions, in order to systematically solve the proposed optimization problem. We can understand the sequential feature acquisition with the following example. As a human doctor diagnoses a patient in the previous healthcare example, we need to decide which unknown features should be discovered in order to be sufficient confident about our prediction. Given a new data point with missing entries, our RL agent sequentially chooses features to acquire based on the set of features acquired so far.

At every examination (or feature acquisition), we pay the pre-defined inspection fee. This process is sequentially repeated until we have collected sufficient but not redundant features to minimize acquisition cost. Once the agent decides to end the acquisition phase, the classifier will make a prediction given the acquired features by the RL agent thus far. At the same time, the final rewards are set according to the prediction made by the classifier, to signal the agent whether the current subset of features is adequate for prediction or not. Interestingly, it turns out that the classifier in our optimization framework can be understood as the estimated environment for the RL agent, which is intuitive in the sense that the reward to the RL agent should be based on how confident our classifier is on its final decision.

It is worth noting that we do not assume a fixed classifier that is pre-trained during a separate phase. Learning the optimal classifier beforehand only by pre-training is unrealistic since the optimal classifier should be associated to the feature acquisition policy; if our acquisition policy changes, the corresponding optimal classifier does as well. Actually, the RL approach for feature selection problem is challenging because it has a huge search space that includes all the variable-sized subset of features and each feature can be either discrete or continuous. To search this huge space efficiently and stably, we carefully devise the architecture with the recently proposed set encoding method [4]. The feature-level set encoding method helps to reduce the state space effectively, by making feature acquisition process to be order-invariant through the attention mechanism [3].

Our contribution is threefold:

- We formulate the feature acquisition problem as an optimization problem with cost-sensitive penalty in the objective function and optimize with a joint framework that simultaneously trains the classifier and the RL agent, for systematic learning of active feature acquisition model, without requiring a probabilistic model or pre-defined classifier.

- We propose a novel method to encode the subset of features by appropriately modifying [4] that can naturally handle missing entries and is shared by the classifier and the agent. In addition, we apply a synchronous variant of n-step Q-learning [5] to handle real-valued feature space in feature acquisition problems.

- We validate the superiority of the proposed framework on diverse simulated and medical datasets and also verify the clinical significance of the results with clinicians, showing that the set of features obtained by our model is similar to what a clinician would use in practice.

## 2   Related Work

As we mentioned in introduction section, we cannot guarantee that not all the features are given away, so we should consider which features to use for testing. First of all, we can select features in static manners, that is, examine same predictors across all instances. It can be done in several different ways including sparsity-inducing regularization [6], greedy forward and backward selections [7].

Instead, we can perform active feature acquisition in more efficient way by considering the differences across instances. Greiner et al. [8] investigate the problem of learning an optimal active classifier based on a variant of the probably-approximately-correct (PAC) model. The following works consider active classification under some constraint. Sheng et al. [9] propose sequential batch test algorithm that acquires a batch of features iteratively until no more positive cost reduction occurs by utilizing a pre-built cost-sensitive model. Kanani et al. [10] identify the data points whose missing features will be completely acquired among all test instances with partially given features, based on the expected utility. Kapoor et al. [11] bridge induction-time and test-time information acquisition in the restrictive case that all attributes and labels are binary. They directly build a statistic model explaining dependency of the test label on its features and also the training data with Bayesian treatment of model parameters. Bilgic et al. [12] approach the problem by constructing graphical models to get the value of subsets of features with the given bayesian network that helps reducing a search space. Trapeznikov et al. [13] consider the budget constraint and suggest a framework to analyze a multi-stage and multi-class sequential decision system such that the $K$ classifiers make a prediction or pass to the next with incrementally acquired features in fixed order. Nan et al. [14] present the classification algorithm which acquires features greedily and sequentially by using partial margin of the k-nearest neighbors defined with the pre-trained linear classifier. Xu et al. [15] construct and train a cost-sensitive classification tree by using trained gradient boosted trees, who inspect different subset of features, as weak learners.

Some existing works formulate the dynamic feature acquisition problem as Markov Decision Process (MDP) or partially observable MDP (POMDP) and try to learn the best feature acquisition policy that gives maximum returns. Ruckstiess et al. [16] set this problem as partially observable MDP (POMDP) with the assumption that pre-learned classifier and prediction result on each instance with missing entries from this classifier as the observation of the state. They solve the POMDP by Fitted Q-Iteration (FQI) but they fail to specify when to stop acquiring features. Dulac-Arnold et al. [17] propose an MDP formulation by directly modeling the state space with acquired features only. They also incorporate the additional special actions corresponding to predictions, which leads to training the classifier (as well as when to stop) implicitly within MDP framework. [18, 19] follow similar MDP setting with ours and they adopt imitation learning approach that train the agent to follow the reference policy that is greedy policy of the oracle who exploit true labels and the classifier. This can restrict the performance of the agent under the oracle's performance that might be suboptimal. Our framework does not assume pre-learned classifier to adapt classifier to the feature acquisition policy and vice versa. Also, our framework learn its policy by directly interacting with and getting reward from the data and the external classifier rather than imitate the suboptimal oracle. Due to the huge search space that is of subset of features, systematic architecture should be designed.

Another recent line of work [20, 21, 22] reduces the computational cost to process the high dimensional image data via the recurrent attention. [20, 21] use REINFORCE [23] to make the models localize informative part and Ba et al. [22] improve these methods with an additional inference network and reweighted wake-sleep algorithm. These models acquire and aggregate observed features by using RNN and after fixed steps of acquisition they make a prediction. While these methods also try to select the most informative subset of features (or area), the goal is different from active feature acquisition: since these models are devised to reduce the computational cost, the number of acquisitions is pre-defined and fixed, depending on the target computational budget, leading to instance-independent feature acquisition costs. Moreover, they do not consider the heterogeneous acquisitions costs across features since all features are in fact always available.

Reinforcement learning that is widely used to find an optimal policy in MDP is also be utilized for information extraction problem that is filling the missing entries by extracting appropriate value from appropriate sources [24, 25]. Kanani et al. [24] consider resource-bounded information extraction (RBIE) as MDP with query, download, and extract as types of actions and learn Q-function by temporal difference Q-learning. Narasimhan et al. [25] address two challenges of utilizing external information that are retrieving suitable external information sources and reconciling extracted entities by employing a deep Q-learning.

## 3 Joint Learning Framework for Sequential Feature Acquisition and the Classification

Consider the standard $K$-class classification problem where we learn a function $f_\theta$ that maps data point $\boldsymbol{x} = (x_1, x_2, \ldots, x_p) \in \mathbb{R}^p$ with $p$ features to a label $y \in \mathcal{Y} := \{1, 2, \ldots, K\}$. The basic assumption here is that the feature vector is fixed-dimensional, and given in full (but possibly with missing entries). We instead consider the same problem under a slightly different condition.

For each data point $\boldsymbol{x}^{(i)}$, we actively acquire the features in a sequential order. Specifically, at $t = 0$ we start with an empty acquired set $\mathcal{O}_0 := \emptyset$. At every time step $t$, we choose the subset of unselected features, $\mathcal{S}_t^{(i)} \subseteq \{1, \ldots, p\} \setminus \mathcal{O}_{t-1}^{(i)}$ and examine the values of missing entries $\mathcal{S}_t^{(i)}$ at the cost $\boldsymbol{c}_t^{(i)} := \sum_{j \in \mathcal{S}_t^{(i)}} c_j$. Hence, after the examination at time $t$, we have access to the values of $\mathcal{O}_t^{(i)} := \mathcal{S}_t^{(i)} \cup \mathcal{O}_{t-1}^{(i)}$. We repeatedly acquire features up to time $T^{(i)}$ ($\mathcal{O}_{T^{(i)}}^{(i)}$ is not necessarily equal for all data points $i = 1, \ldots, n$) and classify $\boldsymbol{x}^{(i)}$ given only the observed features $\mathcal{O}_{T^{(i)}}^{(i)}$. Note that the order of feature acquisitions and corresponding costs can be different across samples, but we suppress the sample index $i$ when it is clear from the context.

In order to learn the model that minimizes the classification loss and the acquisition cost simultaneously, we formulate our framework in the following optimization problem:

$$\underset{\theta, \vartheta}{\text{minimize}} \quad \frac{1}{N} \sum_{i=1}^{N} \mathcal{L}\Big(f_\theta\big(\boldsymbol{x}^{(i)}, \boldsymbol{z}_\vartheta^{(i)}\big), y^{(i)}\Big) + \lambda \boldsymbol{c}^\top \boldsymbol{z}_\vartheta^{(i)} \tag{1}$$

where $\mathcal{L}$ is the cross-entropy loss as in the standard classification problem and the binary vector $\boldsymbol{z}_\vartheta \in \{0,1\}^p$ encodes whether each feature is acquired at the end (or at $T^{(i)}$) when the sequential selection is performed by policy $\vartheta$. Note also that the classifier $f_\theta$ is able to access only available features with $[\boldsymbol{z}_\vartheta]_j = 1$. The hyperparameter $\lambda$ controls the relative importances of the prediction loss and the acquisition cost.

Natural way to solve (1) is to adopt an alternating minimization since the roles and properties of two parameter sets $\theta$ and $\vartheta$ are clearly distinguished in (1). Solving (1) with respect to $\theta_k$ given $\vartheta_{k-1}$ is trivial. Since the cost term $\boldsymbol{c}^\top \boldsymbol{z}_\vartheta^{(i)}$ is a constant w.r.t $\theta$, it is trained to minimize the prediction loss given the subset of features acquired by $\vartheta_{k-1}$. The only challenge here is to find a way of efficiently handing missing features (since the unacquired features by $\vartheta_{k-1}$ are missing to the classifier $\theta_k$).

On the other hand, solving (1) with respect to $\vartheta_k$ given $\theta_{k-1}$ is not trivial. In the following subsection, we will detail how we define a Markov decision process (MDP) to solve (1) w.r.t. $\vartheta$.

## 3.1 RL construction to solve (1) w.r.t. $\vartheta$

Markov decision process (MDP) consists of an agent in the environment, a set of states $S$ and a set of actions $\mathcal{A}$ per state. In state $\boldsymbol{s}_t$, the agent chooses the action according to its policy. Then the state changes and environment gives a reward. The goal of the agent is to maximize the total rewards. In this subsection, we construct an MDP corresponding to solving (1) w.r.t. $\vartheta$ given $\theta_{k-1}$.

**State.**   Since the informative features can be different across classes, the subset of features our RL agent should select will differ across data points. Without having any prior information on true class, the importance of the missing features can be estimated from the currently available features $\mathcal{O}_t$. To this end, we construct the state $\boldsymbol{s}_t$ as the concatenation of $\boldsymbol{z}_t$ and $\boldsymbol{x}_t$ where the $j$-th entry of $\boldsymbol{x}_t$, $[\boldsymbol{x}_t]_j$, is set to zero if $j \notin \mathcal{O}_t$ or to the value of the $j$-th feature otherwise. Note again that $\boldsymbol{z}_t \in \{0,1\}^p$ is the binary indicator encoding which features are acquired until time $t$.

**Action.**   The RL agent selects which features to examine. The set of all possible actions is defined as the power set of $\{1, \ldots, p\}$ including the empty set $\emptyset$, which means to stop acquiring any more features. Throughout the paper, we mainly assume that the agent gets one feature at a time for clarity, hence the size of action space is $p + 1 = |\{1, \ldots, p, \emptyset\}|$. Some actions would be *invalid* if the corresponding features have already been selected previously. $\emptyset$ is a special action that is valid at any time corresponding to 'stop and predict' based on current state $\boldsymbol{s}_t$.

**Reward and environment.**   We naturally define the reward as the negative acquisition cost. Specifically, in the episode $(\boldsymbol{s}_0, a_0, r_1, \boldsymbol{s}_1, \ldots, \boldsymbol{s}_T, a_T = \emptyset, r_{T+1})$, $r_{t+1}$ is set as $-\lambda c_{a_t}$ for $t = 0, \ldots, T-1$. Note that although the state transition from $(\boldsymbol{x}_t, \boldsymbol{z}_t)$ to $(\boldsymbol{x}_{t+1}, \boldsymbol{z}_{t+1})$ is deterministic given $a_t$, it is still not trivial since a new acquisition is unknown before actually observing it.

Unlike 'feature acquisition' actions, the state transition by $\emptyset$ is trivial since no further feature value will be revealed: i.e., $(\boldsymbol{x}_t, \boldsymbol{z}_t) = (\boldsymbol{x}_{t+1}, \boldsymbol{z}_{t+1})$. We also set the final reward $r_{T+1}$ after selecting $\emptyset$ as $-\mathcal{L}\big(f_{\theta_{k-1}}(\boldsymbol{x}_T, \boldsymbol{z}_T), y\big)$ where $\theta_{k-1}$ is given in the alternating minimization scheme.

Then, we have the following result:

**Theorem 1.** *Consider some policy $\pi_{\vartheta_k}$ parameterized by $\vartheta$. Suppose that this policy $\pi_{\vartheta_k}$ is the optimal of Markov decision process described in Section 3.1, given the classifier parameter $\theta_{k-1}$. Then, $\pi_{\vartheta_k}$ is also the optimal solution of (1) with respect to $\vartheta$ given $\theta_{k-1}$.*

The theorem can be simply proven as follows:

$$\operatorname*{argmax}_\vartheta \frac{1}{N} \sum_{i=1}^N \sum_{t=1}^{T_\vartheta^{(i)}+1} r_t\left(\boldsymbol{s}_{t-1}^{(i)}, \pi_\vartheta(\boldsymbol{s}_{t-1}^{(i)})\right) = \operatorname*{argmax}_\vartheta \frac{1}{N} \sum_{i=1}^N \left[ -\mathcal{L}\left(f_{\theta_{k-1}}\left(\boldsymbol{s}_\vartheta^{(i)}\right), y^{(i)}\right) - \lambda \boldsymbol{c}^\top \boldsymbol{z}_\vartheta^{(i)} \right]$$

$$= \operatorname*{argmin}_\vartheta \frac{1}{N} \sum_{i=1}^N \left[ \mathcal{L}\left(f_{\theta_{k-1}}\left(\boldsymbol{s}_\vartheta^{(i)}\right), y^{(i)}\right) + \lambda \boldsymbol{c}^\top \boldsymbol{z}_\vartheta^{(i)} \right]$$

where $\boldsymbol{s}_\vartheta$ and $\boldsymbol{z}_\vartheta$ are respectively the final state and corresponding $\boldsymbol{z}$ by the policy $\pi_\vartheta$. The selection of the final reward above is reasonable in the sense that it should measure how sufficient information for $\theta_{k-1}$ has been provided so far for a prediction.

**Learning policy.** In order to find an optimal policy, we use Q-learning [26] for our agent. Q-learning is a value-based RL method which finds the optimal policy by learning state-action value $Q(s, a)$ from experience. To handle continuous state space, we use deep Q-learning that has shown to be successful in reinforcement learning with high dimensional, high order features and discrete action space $\mathcal{A}$. More specifically, we adopt n-step Q-learning [5] but in synchronous style. [5] makes Q-learning more stable with delayed update of the target network and by running the multiple agents simultaneously to decorrelate the running history. To avoid overestimation of state-action value, we also adopt double Q-learning [27] instead of basic Q-learning.

## 4 Feature-level Set Encoding for Joint Learning

In our framework, we need to learn the state-action value function $Q$ (parameterized by $\vartheta$) and the classifier $C$ (parameterized by $\theta$); note that we rename it from $f_\theta$ to match with $Q$.

Since both components share the input $\boldsymbol{s}$, training them simultaneously can be understood as multi-task learning. Intuitively, $Q$ and $C$ should share certain degree of information between them, since both aims to optimize a single joint learning objective in (1). However, too much sharing could result in reducing the flexibility of each model, and we should find the appropriate level of information sharing between the two. For instance, in case where $Q$ and $C$ are multi-layer perceptrons (MLP), they may share the first few layers. From our preliminary experiment on the effects of information sharing in Figure 1 (see Section 5 for the detail experimental setup), we found out that the partially sharing model outperforms two other extremes (no sharing or complete sharing) in terms of accuracy and the number of observed features, and

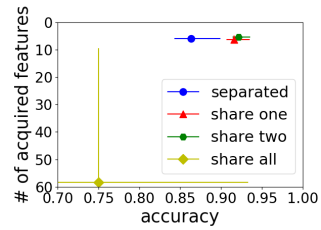

Figure 1: **Effect of sharing layers between $Q$ and $C$:** We check the classification accuracy and average number of features collected on the CUBE with 90 dummy features for varied the number of shared layers from 0 (completely separate) to 3 (completely shared). Both $Q$ and $C$ are MLP with 3 hidden layers whose sizes are 50-30-50. The points and error bars are average and first/third quartile of 100 runs.

also experiences less variance. These shared layers can also be considered as a shared encoder, $\text{Enc}(\boldsymbol{s})$, whose output is fed both into $Q$ and $C$. Hence, our $\text{Enc-}Q\text{-}C$ framework be formulated in the following way. At every time $t$, the state $\boldsymbol{s}_t := (\boldsymbol{x}_t, \boldsymbol{z}_t)$ is fed to the *shared* encoder: $\boldsymbol{h}_t := \text{Enc}(\boldsymbol{s}_t)$. Then, the encoded representation $\boldsymbol{h}_t$ is given to $Q$ and $C$:

$$\boldsymbol{q}_t := Q(\boldsymbol{h}_t), \quad Q(\boldsymbol{s}_t, a) := [\boldsymbol{q}_t]_a \quad \text{for every action } a,$$
$$\boldsymbol{p}_t := C(\boldsymbol{h}_t), \quad \mathbb{P}(y|\boldsymbol{s}_t) := \text{softmax}(\boldsymbol{p}_t).$$

As an example of the feature encoding $\text{Enc}$ in our framework, we devise the feature-level set encoder based on the recently proposed set encoding method [4]. It is composed of two main parts: i) MLP called *reading block*, which maps each set element $x_i$ to the real vector $m_i$ and ii) a *process block* that processes $m_i$ repeatedly with LSTM and the attention mechanism to produce the final set embedding.

We adopt this method to encode each state $\boldsymbol{s}_t$, and individually treat the pair of feature index and its value, $(j : x_j)$, as the set element. We first represent each observed feature as $u_j = (x_j, \mathcal{I}(j))$ where $\mathcal{I}(j) = (0, ..., 0, 1, 0, ..., 0)$ is the one-hot vector with 1 for $j$-th coordinate and 0 elsewhere in order to incorporate the coordinate information. Then, via the set encoding mechanism (through the reading block to make $\{m_j\}_{j \in \mathcal{O}_t}$, followed by the process block) introduced above, we produce the set embedding of the observed features. The *set* encoding is well suited to $\text{Enc}$ since it is invariant according to feature acquisition order and naturally distinguishes two ambiguous cases: $j$-th entry is i) unexplored or ii) discovered but zero value.

### 4.1 Learning and inference

In this subsection, we provide the details on how we can actually train $Q$ and $C$ equipped with the feature-level set embedding $\text{Enc}$ jointly in an end-to-end manner. Due to the space constraint, the pseudocode that summarizes the learning algorithm is deferred to the supplementary material.

We follow the n-step Q-learning procedure described in [5] with two key mechanisms: running multiple agents in parallel and delayed update of target Q-network $Q'$ to prevent perturbation. Specifically, in a training phase, each agent runs for $n$ steps according to its policy based on current $Q$ and gets the experience $(\boldsymbol{s}_t, a_t, r_t, \boldsymbol{s}_{t+1}, a_{t+1}, \ldots, \boldsymbol{s}_{t+n})$. Q-values of invalid actions of each state are manually set to $-\infty$.

After $n$ steps running, $Q$ and $C$ are updated based on the running history. First, the target Q-value $R$ of each state is computed by summing all given rewards after that state plus the approximated Q-value of the last state unless it is the terminal state (with discount factor 1). To avoid overestimation, we use double Q-learning method, so the approximated Q-value is defined as $Q(s_{t+n}, \text{argmax}_a Q(s_{t+n}, a; \vartheta); \vartheta')$ with the target Q network parametrized by $\vartheta'$ that only updated for every $I_{target}$ steps to $\vartheta$ for stability. All parameters of $Q$ are updated by the gradient descent method to minimize the squared error $(Q(\boldsymbol{s}_t, a_t) - R)^2$.

While $Q$ is trained, $C$ is also jointly trained. Since $C$ is supposed to perform a classification task with missing values, it would be helpful to train it with incomplete dataset. Toward this, $C$ is trained on the experienced states that might be incomplete by the gradient descent method to minimize the cross entropy loss: $-\log C_{y_{\text{true}}}(\boldsymbol{s}_t)$ where $C_{y_{\text{true}}}(\boldsymbol{s}_t)$ is the output (or probability after `softmax` layer) corresponding to the true label. $Q$ and $C$ are alternatively updated until the stopping criteria are satisfied. While $Q$ and $C$ have their own loss functions, the shared function `Enc` can be learned both by $Q$ and $C$, or only by one of them, depending on applications.

Before learning $Q$ and $C$ jointly, we pretrain our classifier $C$ with fully observed features and also randomly dropped features with probability 0.5 to simulate random feature acquisition policy. This initialization of $C$ lets joint learning time until convergence be effectively reduced.

**Inference.** At test phase, the start state might be completely empty or partially known. Our RL agent determines which features to acquire by greedily selecting the action with the maximum Q-value until $\emptyset$ is chosen. When $\emptyset$ is selected, $C$ makes a prediction based on the acquired features.

## 5 Experiments

Our code is available at `https://github.com/OpenXAIProject/Joint-AFA-Classification`. To validate the versatility of our model, we perform the extensive experiments on simulated and medical datasets.

**Experimental setup** Throughout all experiments, we use Adam optimizer [28] with 0.001 learning rate and train the models for fixed number of iterations.

### 5.1 Simulated dataset: CUBE-$\sigma$

We first experiment on a synthetic dataset CUBE-$\sigma$ to see if the agent can identify few important features that are relevant to the given classification task. See Fig 2a and [29] for detailed description of the dataset. We train our model on CUBE dataset with 20 features, setting the Gaussian noise $\sigma = 0.1$ for the informative features. We train $Q$, $C$ and `Enc` for 10000 iterations on $10,000$ training instances. It is trained by 4-step Q-learning. Per iteration, 128 agents run in parallel for 4 steps. Instead of updating at once, we do mini-batch update with the size of 128 for 1 epoch (4 times update). We assume uniform acquisition cost $-0.05$ and the final reward as the negative classification loss $-\mathcal{L}$ based on $C$. Both $C$ and $Q$ has two hidden layers of 32-32 units, and `Enc` consists of the MLP with two hidden layers of 32-32 units which maps features to 16 dimensional real-valued vectors and LSTM whose hidden size is 16. For $\epsilon$-greedy exploration, $\epsilon$ linearly decreases from 1 to 0.1 for the first 5000 iterations.

The overall results show that the agent can successfully learn which features are informative and hence it mostly selects the features from the ten informative ones. It consumes only 4.19 features on average while obtaining 96.4% accuracy (even without fine-tuning), which is comparable to the MLP classifier that uses all features (96.98%).

**Effect of Set encoding.** We also evaluate the effect of using proposed feature level set encoding. Given the same structure of $C$ and $Q$, we compare the performances of the proposed set encoding

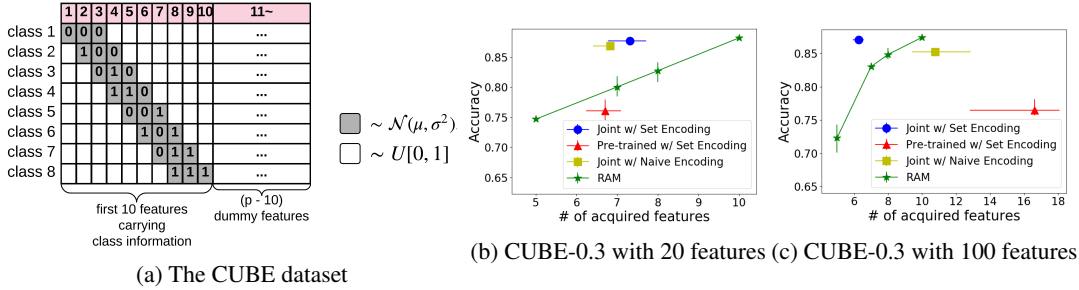

(a) The CUBE dataset      (b) CUBE-0.3 with 20 features (c) CUBE-0.3 with 100 features

Figure 2: (a) The CUBE dataset consists of p-dimensional real valued vectors in 8 classes. The first 10 features only carry class information with three normally distributed entries in the different locations (dimmed in the figure with written mean values) according to the classes. The rest of them are just uniformly random. (b),(c) Comparisons of RAM, our proposed model and its variant on CUBE-0.3. We report averages with 1st and 3rd quantile as error bars of 10 times run.

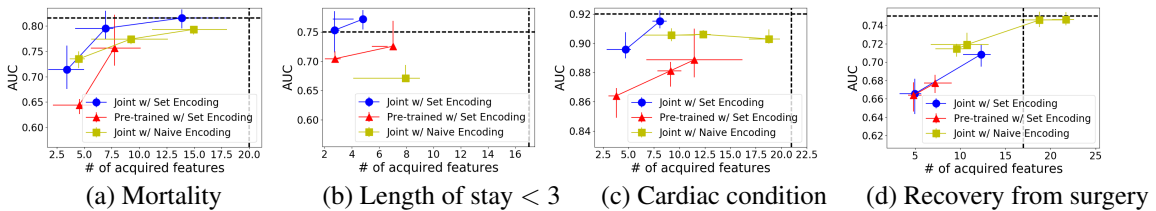

(a) Mortality     (b) Length of stay < 3     (c) Cardiac condition     (d) Recovery from surgery

Figure 3: The comparison on Physionet 2012 data. We report averages with 1st and 3rd quantile as error bars of 10 times run. The black dotted line indicates the result of MLP with group norm.

against the use of naive encoding with shared MLP which takes $(\boldsymbol{x}, \boldsymbol{z})$ as its input. We again use the 20/100-dimensional CUBE data with $\sigma = 0.3$. Fig 2 (b), (c) show the accuracy and the number of features acquired for both encodings. We use cross-validation to select the best setting, and repeat the training ten times with different random seeds. The model with the proposed set encoding outperforms both in terms of accuracy and the number of acquired features, showing the resilience against the increase of dummy features. We conjecture that set encoding can reduce the noise coming from dummy features by attention mechanism so is accompanied by stable policy learning.

Many previous works just assume that the pre-trained classifier is given. However, this is not appropriate for dynamic feature acquisition problem because we need to consider the connection between the feature acquisition policy and the classifier. We compare the jointly learned model with the pre-trained classifier variant in the Fig 2a. We train the classifier with the partial information in the trivial way that simply decides to choose features or not with equal probability and also with the full feature. The result shows that this pre-trained variant has low accuracy with almost the same or more amount of features meaning the classifier cannot perfectly understand the agent's choice.

We additionally analyze how our agent acts on the CUBE dataset and the results are in the supplement.

**Comparison against RAM** Recurrent attention model (RAM) and its variations DRAM, WS_RAM [20, 21, 22] reduce the computational cost while obtaining high performance by taking only the informative parts of an image sequentially. Especially, RAM use REINFORCE to make the models localize informative parts. However, they take glimpses for predefined $n$ steps. It may lead redundant consumption for some cases. We compare RAM on CUBE-0.3 with 20/100 features. The input of RAM is given in the form of $u_j = (x_j, \mathcal{I}(j))$. We can observe in Fig 2(b), (c) that our jointly trained model with set encoding can achieve higher accuracy with less features on average because our agent stops with enough informations and this is not the case for RAM.

## 5.2 Case study: Medical Diagnosis

As our main motivation of this work is to make cost-effective predictions in the medical diagnosis, we further examine how our model operates on real-world medical datasets.

Table 1: Statistics of Physionet 2012 data (# false: # true).

| task \ data | train | validation | test |
|---|---|---|---|
| Mortality | 2586 : 414 | 443 : 57 | 417 : 83 |
| Length of stay $< 3$ | 2917 : 83 | 485 : 15 | 487 : 13 |
| Cardiac condition | 2333 : 667 | 401 : 99 | 392 : 108 |
| Recovery from surgery | 2209 : 791 | 363 : 137 | 360 : 140 |

Table 2: Result on the check-up data. We report prediction AUC (top) and the average number of acquired features (bottom) and cost (USD; in brackets).

| task \ model | Baselines | Joint w/ Naive Encoding | | Pre-trained w/ Set Encoding | | Joint w/ Set Encoding | |
|---|---|---|---|---|---|---|---|
| | MLP | uniform cost | real cost | uniform cost | real cost | uniform cost | real cost |
| Fatty Liver | 0.812 | 0.775 | 0.785 | 0.774 | 0.779 | 0.795 | 0.792 |
| | 35 (577.7) | 6.81 (167.0) | 8.7 (20.1) | 6.2 (127.9) | 4.8 (9.2) | 10.3 (149.8) | 7.5 (13.8) |
| IMT | 0.755 | 0.742 | 0.704 | 0.721 | 0.760 | 0.756 | 0.750 |
| | 22 (1,629.8) | 5.12 (109.9) | 14.22 (11.3) | 3.94 (191.5) | 23.71 (83.8) | 5.57(309.8) | 4.79 (10.0) |

**PhysioNet challenge 2012 dataset**    First, we conduct the experiment on EHR dataset from Physionet challenge 2012 [30]. It has four binary classification tasks, namely in-hospital mortality, whether length-of-stay was less than 3 days, whether the patient had a cardiac condition, and whether the patient was recovering from surgery. We only use the training set whose labels are available and take the features only in the last timestep and split the data randomly into the training/validation/test set by 3000/500/500 ratio. The data is imbalanced (see Table 1), hence we use weighted cross entropy as a loss of $C$.

As baselines, we compare an MLP with group $\ell_1$-norm regularizer for weights in the first layer, DWSC [17], joint training model with naive MLP encoding and the pre-trained classifier variant with our model. DWSC is the model whose action space includes not only feature acquisition but also classification. We omit the result of DWSC in Fig 3 because they often fail to handle the imbalanced data and just learn to predict as the majority. Fig 3 shows the ROC AUC and the number of acquired features. MLP with regularizer also implicitly select features in a static manner. We report the number of nonzero columns of the first layer weight matrix as the number of acquired features shown as a dotted line in the figure. The model in our framework achieves AUC close to that of MLP while acquiring fewer features and outperforms all the others for prediction task of length of stay. This shows that our model is able to select informative, task-related features well.

For in-hospital mortality prediction task, the first feature selected is *Glasgow coma scale (GCS)*, which represents the level of consciousness. GCS is a decisive feature as having a very low GCS means that the patient is almost unsciousness. Thus in such a case, the agent stop the examination and predicts that mortality is true. The other major features are *blood urea nitrogen*, *serum creatine*, *gender* and *age*.

**Medical check-up data**    In order to evaluate our model under a more realistic diagnosis situations where cost comes to play, we perform an additional experiment on check-up data provided by a major hospital. We consider two different binary classification tasks: prediction of 1) fatty liver and 2) intima-media thickness (IMT). We label patents that have steatosis (dB/m) value larger than 280 as having fatty liver (1), and 0 otherwise. For IMT prediction, we label the patients with left or right thickness greater than or equal to 0.8 as positive, and 0 otherwise. Among a total of 3937 data points, steatosis is available for all the data but IMT labels are only available for 1358 instances. Both tasks have 96 features associated with different costs, including body measurement, blood test and CT calcium score, etc. For reality, we consider some features examined together from the one test (such as blood test) as a one multidimensional feature and so make them achieved at the same time. We randomly split the data into three folds with the ratio of 64:16:20 for train:validation:test.

We first compare the AUC to the MLPs with group $\ell_1$-norm regularizer on the weight of the first layer to leverage static feature selection effect. We here report the results selected by cross validation. Since we are concerned with cost-sensitive feature acquisition in this experiment, we report not only the average number of examined features but also the average examine cost. We also compare two different cost settings: 1) A uniform constant across all the features and 2) Actual examine cost

multiplied by a constant $\lambda$. Examination costs of the features determining the labels are about 80 and 1,564 USD for fatty liver and IMT respectively, and thus our aim is to make diagnosis with lower costs. As shown in Table 2, an MLP and our model both achieve comparable AUC, while our model use significantly less cost (13.8 USD for fatty liver and 10 USD for IMT).

Now, we briefly describe the detail of diagnosis process of our model. The followings are verified by the clinicians. MLP incurs large examination cost because it leverages coronary calcium score and Echo E/E' for prediction, and so does our model with uniform cost setting as it selects Echo EF value which are all expensive. However, our cost-sensitive model effectively acquires relatively cheaper and relevant features for fatty liver prediction such as age, BMI, low-density lipoprotein (LDL), etc. Similarly in the IMT prediction task, rather than taking expensive features such as Echo E/E' and steatosis (it is chosen because IMT is known as to be related to fatty liver disease) as done by MLP or our model with uniform cost, our cost-sensitive model makes a decision based on age, smoking history, genetic factors, weight, BMI (obesity), glucose (diabetes), blood pressure, LDL cholesterol, TG (cholesterol). This result is meaningful because all these are well-known risk factors of atherosclerosis that are also often used by doctors.

## 6 Conclusion

A cost-aware sequential feature selection can be used in situations where the features are not provided in full and each collection of features incurs variable cost, such as with medical examination. To solve this problem, we formulated it into an optimization problem of simultaneously minimizing the prediction loss and the feature acquisition costs, and derived a joint learning framework for the classifier and the RL agent. We validated our model on both synthetic and real medical datasets to confirm the superiority of proposed feature level set embedding under our joint learning framework.

However, there is still room for improvement of our model. First of all, in the synthetic dataset experiment, we found that features are overly acquired for a few instances whose class is indistinguishable because of the data generation rule. This problem may be due to the intrinsic overestimation behavior of Q-learning because Q-learning update uses maximum approximated action value as a target. Therefore, Q-values of feature acquisition actions which are nonterminal are overestimated compared to stop action. We plan to explore some tricks or other reinforcement learning methods to alleviate this problem as future work. Secondly, we can take model uncertainty into account, so that our agent collects the features that can help to reduce uncertainty as well as increase classification performance with as small as possible amounts of cost.

When it comes to running time, training time is less than 20 minutes for reported experiments, and for testing, it takes about 0.5 sec to evaluate 500 instances (on GTX 1070). In training phase, convergence is affected by decaying schedule for $\epsilon$ of $\epsilon$-greedy policy in Q-learning algorithm. We found that decaying it linearly from 1 to 0.1 from the first half of total iterations generally works well.

## 7 Acknowledgements

This work was supported by Institute for Information &communications Technology Promotion(IITP) grant (2017-0-01779, A machine learning and statistical inference framework for explainable artificial intelligence, 2017-0-00537, Development of Autonomous IoT Collaboration Framework for Space Intelligence), the Engineering Research Center Program through the National Research Foundation of Korea (NRF) (NRF-2018R1A5A1059921) funded by the Korea government(MSIT) and NAVER Corporation (Research on deep and structurally constrained machine learning).

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
