[Supplementary Material]



(a) Case 1        (b) Case 2        (c) Learned policy on different costs

Figure 4: **(a-b) Attentions on features in two example cases.** $x$ and $y$ axes represent the feature index and the attention value respectively. Only informative features for classification are being focused on for both cases. **(c) Learned policy on different costs features.** Experiment with two features that provide the same information at different costs. The bar plots denote the number of times each feature is acquired.

# Appendix

## A  Detail analysis on CUBE

**data generation**   The name "CUBE" came from the fact that each data point is assigned to one of eight corners of length 1 cube and $\sigma$ represent the gaussian noise added on each coordinate. Specifically, it is generated by the following rules. first, each value of three features $(x, y, z)$ is set to 0 or 1 with same probability. Class 1 is correspond to $(0,0,0)$, $(1,0,0)$ is class 2, $(1,1,1)$ is class 8 and so on. Gaussian noise $\epsilon \sim \mathcal{N}(0, \sigma^2)$ are added to these three values. Note that the three information-carrying features are not located in the same position but they are shifted over classes. For class $i = 1, 2, \ldots, 8$, $x$ is positioned on $i$-th feature and $y, z$ are followed. The remaining features are filled with the values sampled from uniform distribution between 0 and 1. Thus, among $p$ features first ten are informative and the rest are dummy features. We used $\sigma = 0.1, 0.3$. On the cube with 20 features, as $\sigma$ increasing from 0.1 to 0.3, the accuracy of the MLP classifier decreases from 96-97% to about 86%.

**Analysis of Episodes.**   We take a closer look at how efficiently our trained model works by examining two specific test episodes summarized in Table 3. The first episode, where the true class is 8, consists of 3 feature acquisition actions. Following the greedy policy, the agent collects 7th, 3rd and 5th features in an order and makes a prediction. The first acquired 7th feature (0.855) informs the model that the class is unlikely to be 6 or 7 whose $\mu = 0$ (see Figure 2a). Next, by acquiring the third feature (0.667), that is highly likely to be generated from the uniform distribution, class 1,2 and 3 are excluded. In terms of the certainty of classifier, the probability of true class 8 increases from 0.16 to 0.37 after step 2. The next greedy action is to select the 5th feature (0.6796), which allows the agent to further eliminate class 4 and 5. After these 3 steps, the agent stops, and the classifier gives the answer as 'class 8' with probability 0.99, which means that it is almost certain about its this prediction. The second episode in Table 3 is more extreme. The agent acquires the 7th feature that is greater than 1. In this case, the agent is certain that the value is from the normal distribution since it is beyond 1. The only possible answer here is 'class 5' where 7th feature is generated from the normal with $\mu = 1$. Our agent successfully catches this and predicts the class only after a single feature collection.

**Analysis of Attention.**   The encoder `Enc` uses the attention mechanism in the process block. In order to examine how it works, we select two typical examples among the cases where the agent observe more than 10 features. In the first case (Figure 4(a)), the agent observe all 20 features. Note that this occasionally happens since some data point indeed is quite difficult to be distinguished due to the overlap between uniform and Gaussian distributions (for instance, suppose we have some values close to $(0, 1, 1, 1)$ for 7th-10th features. Then, it is indistinguishable for class 7 or 8). In this case, the agent tends to collects more features, since $\emptyset$ will give low reward in high probability right away while $Q$ overestimates the value of the feature acquisition actions. As shown in Figure 4(a), the attention focuses only on the task-related features (3rd to 10th). In the second case (Figure 4(b)), the agent obtains the first 10 features for class 7 data. To differentiate the class 6, 7, and 8, we need features from 6th to 10th except the 8th feature that are sampled from $\mathcal{N}(1, 0.1^2)$ for all three classes (see Figure 2a). We observe that our model allocates attention to only those four important features.

Table 3: Example episodes to show how our trained agent acts on the cube dataset. The agent successfully selects the cost-effective subset for both two example cases.

| | | First episode ($c = 8$) | | | Second Episode ($c = 5$) | |
|---|---|---|---|---|---|---|
| | Init state | Step 1 | Step 2 | Step 3 | Init state | Step 1 |
| Selected feature | | 7th | 3rd | 5th | | 7th |
| Feature value | | 0.855 | 0.667 | 0.6796 | | 1.146 |
| $\mathbb{P}(y = c\|s_t)$ | 0.123 | 0.169 | 0.373 | 0.995 | 0.121 | 0.985 |

**What if the feature acquisition costs are not uniform?** So far, we showed that our model collects the informative but not redundant features. We further examine the case that the acquisition costs of the features are not uniform. Toward this, the generation rule of CUBE-0.3 is slightly changed so that 11th to 20th features are generated in the same way of the first 10 following 80 dummy features ($i/(i + 10)$-th ($i = 1, \ldots, 10$) features give almost the same information). We then assign the uniformly random cost in the interval $(0.05, 0.1)$ for feature acquisition. The difference of the cost between $i/(i + 10)$-th features are from 0.002 to 0.04. As we expected, the cheaper feature are preferred to the other (see Fig 4(c)). Our model gets 0.92 accuracy while observing 5.62 features on average.

# B Pseudocode

---
**Algorithm 1** Learning process of our framework

---
**Input:** training dataset $(X, y)$ with full information
Ramdomly initialize model parameters $\theta$ of $C$, $\vartheta$ of $Q$
Initialize target network weights $\vartheta' \leftarrow \vartheta$
Pretrain $C$ with full and randomly dropped features with probability 0.5
Initialize exploration
**repeat**
  // Run $N$ agents for mini-batch $\{x^{(i)}\}$ in parallel
  $t_{start} \leftarrow t$, get state $s_t = (x, z_t)$
  **repeat**
    Take action $a_t$ according to the $\epsilon$-greedy based on $Q(s_t, a; \vartheta)$
    **if** $a_t$ is feature acquisition action **then**
      Receive $r_t = -\lambda c_{a_t}$ and feature value $x_{a_t}$
      $m_t \leftarrow 1$
    **else**
      Receive $r_t = -\mathcal{L}(f_\theta(s_t), y)$
      $m_t \leftarrow 0$, sample new state $s_{t+1} = (x, 0)$
    **end if**
    $t \leftarrow t + 1$
  **until** $t - t_{start} == n$
  $R \leftarrow Q(s_t, \text{argmax}_a Q(s_t, a; \vartheta); \vartheta')$ (double DQN)
  **for** $j \in \{t - 1, \ldots, t_{start}\}$ **do**
    $R \leftarrow r_j + m_t R$
    $d\vartheta \leftarrow d\vartheta + \frac{\partial (R - Q(s_j, a_j; \vartheta))^2}{\partial \vartheta}$
    $d\theta \leftarrow d\theta + \frac{\partial \mathcal{L}(f_\theta(s_j), y)}{\partial \theta}$
  **end for**
  update $\vartheta$ with $d\vartheta$ and then update $\theta$ with $d\theta$
  **if** $t \bmod I_{target} == 0$ **then**
    $\vartheta' \leftarrow \vartheta$
  **end if**
**until** $t == t_{max}$

---

## C Comparison against baselines

We compared our method against two traditional feature-selection baselines during the rebuttal period: (1) LASSO (2) Expected utility (EU), where utility is defined as a difference of class probability entropy per cost. LASSO conducts static feature selection by inducing sparsity of the parameters. For expected utility method, we use the same size classifier with our model for fairness. Utility is defined as the difference of Shannon entropy of classification probability before and after accessing to features. For the formal definition of expected utility, we refer the readers to [10]. Because we do not know the true distribution of missing features, we construct a deep neural network to estimate density of each unknown feature for imputation. For simplicity, we assume the distribution of $i$-th unknown feature $x_i$ given current state $s_t$ as Gaussian distribution $P(x_i|s_t) = \mathcal{N}(x_i|\mu_i(s_t), \sigma_i(s_t)^2)$. To train $p$ imputation networks, we sample $s_t$ from our dataset and take all features or drop them randomly with the probability 0.5. We optimize $\mathcal{L}_i(\theta_i) = \frac{1}{N}\sum_{j=1}^{N} \frac{1}{2\sigma(s_t^{(j)})^2}(x_i^{(j)} - \mu(s_t^{(j)})^2) + \frac{1}{2}\log\sigma(s_t^{(j)})^2$ as [31] did to model heteroscedastic aleatoric uncertainty where N is mini-batch size. With the pre-trained imputation networks and the classifier, we conduct sequential feature acquisition based on the expected utility score, until that score becomes negative. Expected utility is approximated by Monte Carlo method, that is, averaging utilities evaluated with samples of the unknown feature from the Gaussian distribution obtained by imputation networks. The results in Table 4 show that our method significantly outperforms the simple baselines in classification AUC (accuracy for CUBE), while using less number of features on average except on 'mortality' task on which EU performs similarly. (Note that we use the input normalization for this experiment since LASSO severely underperforms otherwise).

Table 4: Comparison against traditional feature-selection baselines (numbers in brackets represent the number of collected features)

| Dataset | IMT | CUBE | Mortality | Length of stay $< 3$ | Cardiac condition | Recovery from surgery |
|---------|-----|------|-----------|----------------------|-------------------|-----------------------|
| Ours | **0.76(5.6)** | **0.87(6.4)** | 0.83 (9.6) | **0.74(6.8)** | **0.92(8.1)** | **0.74(9.5)** |
| LASSO | 0.73 (6.0) | 0.53 (7.0) | 0.77 (10.0) | 0.69 (10.0) | 0.88 (10.0) | 0.72 (10.0) |
| EU | 0.68 (7.7) | 0.81 (6.9) | **0.84(9.7)** | 0.60 (7.0) | 0.89 (10.6) | 0.72 (13.0) |