[Reviews · NeurIPS 2018]

Reviewer 1



The authors consider the problem of active feature acquisition (i.e., the setting where each feature includes a cost to see the value, often motivated by diagnostic tests in medical settings), where the goal is to maximize classifier performance with a minimum cost subset of features. Methodologically, they start by modifying the objective function to include empirical risk minimization and feature acquisition cost minimization. The acquisition mechanism is done via a RL formulation in a method similar to RL/imitation learning methods for feature selection (e.g., [15-18]), where the state is the observable features, the action is to acquire a new feature (or classify), and the reward is the (negative) feature acquisition cost. The interesting aspect of their approach is that the classifier and RL agent are jointly optimized (RL via deep Q-learning), requiring a clever set encoding method to manage the large search space (similar to [Vinyals, Bengio & Kudlur; ICLR 2016]). The joint optimization is completed via a joint encoding of the classification and policy with shared encoder layers. Experiments are first conducted on synthetic data sets to elucidate specific properties (e.g., the value of joint training vs. a pre-trained classifier, the benefits of set encoding). Secondly, experiments are conducted on PhysioNet (four classification tasks) and a private Medical check-up dataset. Evaluating using AUC, they show that they are able to generate competitive results while observing ~20% of the features. In summary, this is an interesting approach to feature value acquisition that builds on recent work wrt meta-learning with RL-based approaches and adopting this work to this particular setting with the joint learning and set encoding methods that are entirely sensible. Additionally, they provide sufficient training details where it seems likely that others would be able to produce this work. Experiments are conducted on synthetic sets, ‘real’ (bake-off) datasets, and an actual real-world problem. The only technical detail I find a bit surprising is that they are able to train these algorithms with 1000s of examples, when policy gradient is generally fairly data hungry, etc. Thus it would also be nice to see more ‘traditional’ baseline results (i.e., no deep learning, just an expected gain test) — particularly for the real-world problems. While I am not a true expert on this particular problem, this is ostensibly a novel approach to a potentially important problem with reasonably convincing results. However, the writing needs significant work (and I am not entirely certain it followed the latest style files) including structurally, grammatically, and even some of the precision of claims (covered in more detail below). Accordingly, I am ambivalent wrt accepting this work in its current form, learning toward rejection given the likely number of strong papers submitted. [Quality]: Overall, the paper is well-motivated, specifies a sensible model, provides a reasonable level of implementation details, and generates promising experiments. While the writing needs work, I will leave this to ‘clarity’. I do have a few questions that are a bit nit-picky: (1) the authors state a ‘cost-sensitively (sic) regularized optimization problem’; however, it isn’t clear to me that this is a regularizer per se as I generally think of regularization as a variance reduction mechanism — not all penalty functions within the objective will do this (and I’m not certain this does), (2) the authors make a point that ‘To search this huge space efficiently and stably, we carefully devise the architecture with the recently proposed set encoding method’ — as best as I can tell, this is exactly the same method, correct?, (3) Lines 131-132: “the hyperparameter balancing the relative importances of two terms…is absorbed in the predefined cost” — isn’t this difficult to do while keeping the costs interpretable (or is this just notational)?. Overall, precisely clarifying these differences wrt existing work would strengthen the paper. [Clarity]: The paper needs significant writing work. I will give examples of each issue (in my opinion), but it is an overall issue. - Structure: The introduction starts with a discussion around deep learning and proceeds to feature acquisition — it isn’t entirely clear why deep learning is a significant aspect of this paper. I would think one would start with the feature acquisition aspect and then proceed to deep learning (although I think it really is secondary). The related work section is basically a ‘laundry list’ of papers. Without reading them, they are mostly just citations — while this is an opportunity to provide a general framework and demonstrate how this fits together. - Grammar: The paper is full of simple errors like missing determiners, syntactic errors, odd word choice, tc. (i.e., Line 37: “Given a new dataset”; Line 36: “to be sufficient confident”; Line 47: “It is worth noting that we do not assume a fixed classifier that is pre-trained during a separate phase.; Line 57: “We formulate the feature acquisition problem as a cost-sensitive penalty in the objective function and optimize with a joint…”; Line: 69: “…we cannot guarantee that all of the features are available at the start”; Line 98: “Also, our”). These are all over the place. Finally, I think there should be more discussion around the empirical results in general. [Originality]: While I am not a true expert in this area, while it seems to draw significantly from related work, it also put together multiple relevant works and ostensibly thought through the details — showing promising results. While I think many could have come up with this formulation, I also don’t believe it trivial and is an advance over more directly formulations. [Significance]: While I would like to see additional experiments contrasting with non-deep learning methods, this work does seems like a promising approach to active feature value acquisition. This is a practical problem (even if somewhat narrow in scope) — so I would say that it meets the bar in this regard. Overall, this paper has good ideas, solid execution, and promising results. However, given the competitiveness of NIPS, I think the paper need significant writing work to be competitive in its current form. Comments after author rebuttal: I do appreciate Table A and in reading the other reviewers comments, I may have over-indexed on the writing (which does need work, but it is difficult for me to validate) and under-appreciated the effort to get this validated in a practical setting. Thus, I am a bit more positive wrt this work, but still believe it should be somewhat borderline and wish we had more confident reviewers.

Reviewer 2



This paper proposes a deep-RL based approach for jointly learning a classifier along with an active feature acquisition strategy. Overall, this is a relatively less studied area of ML, and this work is a good step towards building a general solution. Major strengths: - Overall learning approach seems sound, interesting, and novel - Use of feature-level set encoding is an interesting application of some recent work to this area - Experiments on both synthetic and real world datasets, including attempt to get it verified by domain experts (in this case medical professionals) Major weaknesses: - Some missing references and somewhat weak baseline comparisons (see below) - Writing style needs some improvement, although, it is overall well written and easy to understand. Technical comments and questions: - The idea of active feature acquisition, especially in the medical domain was studied early on by Ashish Kapoor and Eric Horvitz. See https://www.microsoft.com/en-us/research/wp-content/uploads/2016/12/NIPS2009.pdf There is also a number of missing citations to work on using MDPs for acquiring information from external sources. Kanani et al, WSDM 2012, Narsimhan et al, "Improving Information Extraction by Acquiring External Evidence with Reinforcement Learning", and others. - section 3, line 131: "hyperparameter balancing the relative importances of two terms is absorbed in the predefined cost". How is this done? The predefined cost could be externally defined, so it's not clear how these two things interact. - section 3.1, line 143" "Then the state changes and environment gives a reward". This is not true of standard MDP formulations. You may not get a reward after each action, but this makes it sound like that. Also, line 154, it's not clear if each action is a single feature or the power set. Maybe make the description more clear. - The biggest weakness of the paper is that it does not compare to simple feature acquisition baselines like expected utility or some such measure to prove the effectiveness of the proposed approach. Writing style and other issues: - Line 207: I didn't find the pseudo code in the supplementary material - The results are somewhat difficult to read. It would be nice to have a more cleaner representation of results in figures 1 and 2. - Line 289: You should still include results of DWSC if it's a reasonable baseline - Line 319: your dollar numbers in the table don't match! - The paper will become more readable by fixing simple style issues like excessive use of "the" (I personally still struggle with this problem), or other grammar issues. I'll try and list most of the fixes here. 4: joint 29: only noise 47: It is worth noting that 48: pre-training is unrealistic 50: optimal learning policy 69: we cannot guarantee 70: manners meaning that => manner, that is, 86: work 123: for all data points 145: we construct an MDP (hopefully, it will be proper, so no need to mention that) 154: we assume that 174: learning is a value-based 175: from experience. To handle continuous state space, we use deep-Q learning (remove three the's) 176: has shown 180: instead of basic Q-learning 184: understood as multi-task learning 186: aim to optimize a single 208: We follow the n-step 231: versatility (?), we perform extensive 233: we use Adam optimizer 242: We assume uniform acquisition cost 245: LSTM 289: not only feature acquisition but also classification. 310: datasets 316: examination cost?

Reviewer 3



The paper proposes a feature selection/acquisition method to achieve better predction performance. The authors present a novel reinforcement learning framework for training the RL agent to select new feature or to predict. In general, the idea of jointly minimizing the prediction error and feature acquisition cost is clear and convincing. The experimental results show that the performance of the proposed method is favorable. Nevertheless, there are still some concerns to be addressed. 1, Novelty. One of the contributions of this paper to the literature is the joint training framework for learning active feature acquisition model. I am wondering if they are the first one who use the RL agent for selecting features without a pre-defined classifier. The proposed feature-level set encoding for sharing information between the learning tasks is designed as shared layers, and the set encoding method is from [4]. So, why the claimed encoding method is novel in this paper? 2, Presentation. The paper reads well overall, although some descriptions are not clear enough. For example, it's better to use the full name of reinforcement learning instead of its abbreviation RL in abstract. In Section 3, the basic assumption here is that the feature vector is fixed-dimensional, and given in full (but possibly with missing entries), what do you mean missing entries? Does this mean some values of one feature vector are not available? 3, Experiments. The efficiency of the algorithm is also important for the task, what is the running time of the proposed method and its components. The analysis of the algorithm limitations is encouraging, visual examples may need to support the authors’ arguments.